# Evaluation and Optimization of Green Space Fairness in Urban Built-Up Areas Based on an Improved Supply and Demand Model: A Case Study of Chengdu, China

Qidi Dong [1],* 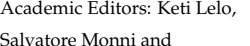, Heng Lu [2], Xiaohong Luo [3], Pengman He [4], Di Li [5], Linjia Wu [6], Yundi Wei [1] and Xuli Chen [1]

1   School of Art and Design, Xihua University, Chengdu 610039, China; 0720180001@mail.xhu.edu.cn (Y.W.); chenxuli@mail.xhu.edu.cn (X.C.)
2   Sichuan Academy of Forestry, Chengdu 610036, China; luhe9212@gmail.com
3   Chengdu Park City Construction and Development Research Institute, Chengdu 610036, China; lorrylxh@gmail.com
4   College of Landscape Architecture, Sichuan Agricultural University, Chengdu 611130, China; hpm@stu.sicau.edu.cn
5   Geophysical Exploration Brigade of Hubei Geological Bureau, Wuhan 430100, China; lidi@stu.sicau.edu.cn
6   College of Art, Sichuan Tourism University, Chengdu 610100, China; wulinjia@sctu.edu.cn
*   Correspondence: dqd@mail.xhu.edu.cn

**Abstract:** Urban green space (UGS) is an important public infrastructure. However, the rapid development of cities and the insufficient supply and uneven distribution of UGSs have led to a mismatch between them and various needs, which has seriously affected environmental justice and social equity. This study discusses the fairness of UGS from the perspective of supply and demand and improves the theoretical framework of supply and demand combination evaluation. First, this study analyzes the supply of UGSs through accessibility analysis and evaluates their demand from both subjective and objective dimensions. Second, the fairness of UGS is analyzed from a Gini coefficient and spatial evaluation perspective. Finally, the location optimization of UGSs is performed, which provides strategic guidance for the improvement of UGSs in Chengdu. The research results based on Chengdu City show that the travel mode directly affects the supply range of UGSs and is reflected in their fairness. At the same time, in the case of a highly dense population, UGSs in the city are in short supply; that is, the equity is negatively correlated with the population. This study provides a new perspective to evaluate UGS fairness and can be a reference for UGS optimization decisions.

**Keywords:** urban green space; big data; supply–demand perspective; urban spatial equity; spatial optimization

## 1. Introduction

Urbanization is an important issue in global development. In 2016, the United Nations put forward the Sustainable Development Goals, which include Sustainable Cities and Conventions, Good Health, and Well-being [1]. In the process of urbanization, urban green space (UGS), as an important urban public infrastructure, is an important place to ensure the sustainable development of the city and the quality of life of urban residents. However, in the early period of rapid urbanization and rapid economic and population growth, urban ecosystems were irreversibly affected [2]. At the same time, the enormous contradiction in the relationship between resource supply and demand limits the equity of green space resources for residents [3]. In the past, UGSs were mostly used as appendages of urban construction. Traditional indicators such as per capita green space and green coverage rate are commonly used to evaluate the quality of UGSs, and the method of "filling in gaps" is adopted to perform UGS construction. As a result, the contradiction between the supply and demand of UGSs has become increasingly prominent, which greatly affects social

fairness and residents' satisfaction and is not conducive to the sustainable development of cities. Therefore, it is necessary to develop a more scientific method of equality research that promotes the equal use of urban infrastructure by urban residents and contributes to the promotion of the health and well-being of urban residents.

Urban equity mainly goes through three stages: regional equity (equal distribution of public facilities), spatial equity (utilization efficiency of land resources), and social equity (fairness to people). Accessibility, as a quantitative research method of spatial equity, analyzes urban public facilities from the perspective of social equality and humanization, and relevant studies often take this as the basis of equity [4]. As early as 1959, Hansen proposed the concept of accessibility, indicating the difficulty of getting from one place to another [5]. Many scholars have discussed the fairness index of UGS from different angles. Handy S L et al. combined distance attenuation theory with the gravity model to provide a theoretical framework for the measurement of accessibility [6]. In terms of the factors that influence accessibility, studies have proven that large UGSs are visited more than small UGSs, and the accessibility level is higher [7–9]. Moreover, some scholars believe that the number, location, distance, area, shape, and other UGS-related factors have varying degrees of influence on UGS accessibility [10–12]. However, with the increasing complexity of urban road network types [13,14] and user needs [15], it has become increasingly difficult to analyze UGS accessibility and the corresponding influencing factors in detail. The emergence of geospatial big data can make up for this shortcoming to some extent. Martin D et al. developed software for analyzing designated bus travel by using bus schedule data [16]. Some researchers have combined modes of transportation (e.g., cars, public transit, pedestrians, and bicyclists), comprehensively considered the accessibility of UGS to different modes of transportation, and included differences in transportation choices for different groups of people [16,17]. Salonen M et al. optimized accessibility based on three methods used to calculate travel time and an API provided on the website of a travel itinerary designer [18]. Du et al. used the AmAP API to assess the travel modes of walking, cycling, and public transportation and calculated the travel times from residential areas to UGSs; the results were used to help balance UGS supply and demand [19]. Existing accessibility studies have established a relatively complete evaluation system on the supply side of UGS, but it lacks classification according to road quality or focus on a single travel mode dimension, resulting in evaluation results that are inconsistent with the actual situation of the city.

With the deepening research on the fairness of UGSs, the fairness of UGSs cannot be fully explained from the supply side of single accessibility, and the explanation of the service availability of UGSs is also weak [20]. At present, only a few articles have studied the fairness of UGSs from the perspective of supply and demand. Among them, Xu, Yu, and others have relied on population indicators to measure the degree of matching between population density and UGS at the urban or regional scale. The results showed that the supply and demand matching of UGSs showed a high positive correlation with the impact of population size and population density, indicating that residents in areas with high population density enjoyed fewer green space resources [21,22]. Xing and Liu et al. further analyzed the compliance of UGS supply and demand based on indicators such as green space area, proportion of green space, and per capita green space [23,24]. Relying on the quantitative data of recreation, ecological regulation, and carbon footprint provided by UGS, some scientists have proposed a key assessment system to determine the supply and demand of UGS [25,26]. In other aspects, small-scale studies on the relationship between the supply and demand of UGSs mostly start from the main body of residents and rely on the satisfaction evaluation of UGSs to analyze the matching degree of supply and demand [27]. Relevant studies have analyzed the matching degree of supply and demand from the perspective of residents from the aspects of function, activity, concern, preference, and cognition [28–30]. Rahman K found that although UGS provided the functions required by users, such as recreation, environmental aesthetics, and social interaction, it could not meet the educational service functions required by users [31].

However, the current research on UGS fairness still has the following problems. First, on the supply side of UGSs, existing accessibility studies have established a relatively complete evaluation system for the UGS supply side, but there is a lack of classification of road quality and a focus on single-dimensional travel modes, resulting in evaluation results that do not conform to the real situation of the city [32]. Second, on the demand side of UGS, most studies focus on the subjective level. Relevant studies generally discuss the use needs of urban residents for UGSs at the scale of parks/communities but lack an analysis of urban/regional spatial development demand. As a result, most studies focus on small and medium-sized scales, which cannot be matched with urban/regional studies in supply, which is also one of the reasons for the lack of supply and demand studies [33–35]. Overall, the current research on UGS supply and demand is relatively independent in terms of research object, scale, and evaluation content, and it is difficult to directly analyze the relationship and matching between the two, so the guiding role of UGS planning, construction, and optimization is not obvious.

In this study, we aim to analyze the fairness of UGS from the perspective of supply and demand. In view of the mismatch between the supply and demand of UGSs, the core goal of this study is to explore multidimensional research ideas and methods of UGS equity that are more in line with the actual situation of the city. We select open data with geo-information properties in big data [18,19], such as Urban traffic data, Point of interest (POI) data, etc., as the basis, and complete the quantification and superposition of data through accessibility analysis, demand analysis, and other methods. This study provides a theoretical basis and technical support for the rationality and humanism of UGS planning. The following research questions guide our research:

What are the contents and indicators of UGS fairness evaluation from the perspective of supply and demand?

How can the unity of supply and demand in the research object, scale, and evaluation content be achieved?

How can supply and demand be matched and the fairness of UGSs be analyzed?

## 2. Materials and Methods

### 2.1. Methodological Design Flow

Figure 1 shows the detailed workflow of this study (Figure 1), which includes three parts: supply and demand evaluation, fairness evaluation, and optimization.

Based on the improved supply–demand perspective, this study explores the fairness of UGS in urban built-up areas. After data selection and processing, supply and demand are evaluated separately. On the supply side, the network analysis database of multidimensional travel modes is established, and the reachability is measured by means of weighted superposition. Second, the evaluation system combining "subjective–objective" is put forward in the aspect of demand. In the second part, based on the evaluation results of the previous step, the service dead zone and imbalance areas of UGS service were identified, and the UGS fairness evaluation from the perspective of supply and demand was completed by means of superposition. In addition, the Lorenz curve and Gini coefficient were combined to understand the status quo of regional fairness from the data. In addition, the Lorenz curve and Gini coefficient were combined to understand the status quo of regional fairness from the data. In the third part, based on the location assignment model, the points for optimizing location selection by UGSs are selected, and the steps in the second part are repeated to compare the fairness of UGSs before and after optimization.

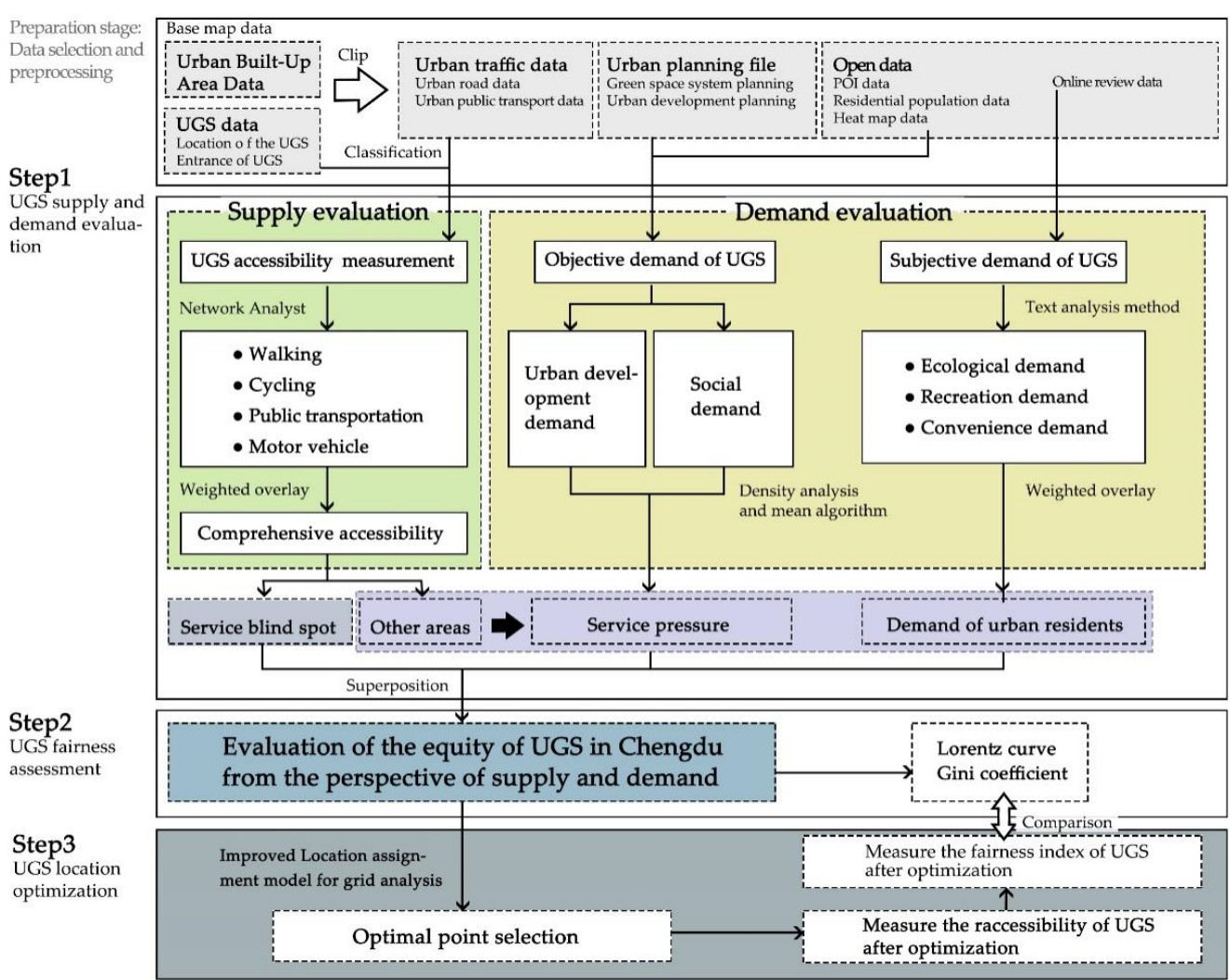

**Figure 1.** Diagram of the proposed method and processes.

*2.2. Study Area*

Chengdu, the capital of Sichuan Province, China, is a high-density city with a permanent urban population of more than 21 million. With its unique natural landscape resources and climatic conditions, Chengdu is known as the "Land of Abundance". There are two reasons for our choosing of Chengdu as the study area. First, research on UGS accessibility in China mainly focuses on first-tier cities such as Beijing and Shanghai. After a long period of development, these first-tier cities have formed a relatively complete UGS system. The development of UGSs in second-tier cities such as Chengdu is still at a low level, and there is a large gap between second-tier cities and first-tier cities, which is worth further research and consideration. Second, since the 1990s, China has proposed standards and titles for evaluating urban human settlements such as "garden cities" and "forest cities", which have greatly boosted the construction of UGSs. In 2019, "park city" became the focus of the development of human settlements in China today. Chengdu is the first city in China to propose the construction of a "park city". At present, the green coverage rate of Chengdu has reached 42.47%, but there is a large gap with the requirements of "green at 300 m and garden at 500 m" in future urban planning [36]. Therefore, the analysis of the fairness of UGSs in Chengdu is of great practical significance for understanding the current situation of urban UGS supply and demand and promoting the construction of "park cities". In this study, we chose UGSs in the urban built-up area of Chengdu as the research object, focusing on their allocation equity (Figure 2).

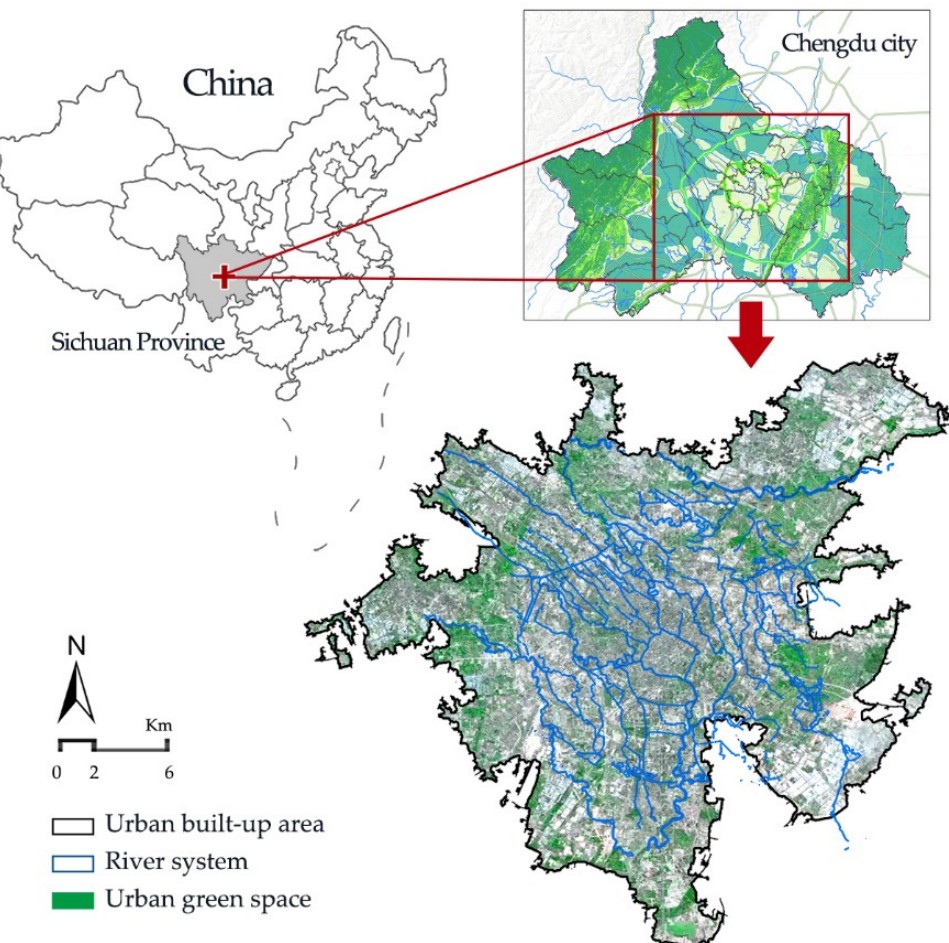

**Figure 2.** Overview of the study area.

*2.3. Data Sources and Pretreatment*

2.3.1. Urban Built-Up Area Data

Urban built-up areas are highly intensive areas of urban construction land and buildings, reflecting the size and density of buildings [37]. This content is an important basis for defining the boundaries of our research and completing data processing We refer to the 2020 China urban built-up area dataset shared by Li et al. in the Science Data Bank (https://www.scidb.cn/ (accessed on 3 January 2023)) [38]. The data are based on global high-resolution artificial impervious area data (longitudinal scale of 30 m), which can effectively depict the extent of built-up areas in the city. Using these data, we identified the built-up area of Chengdu as 1561.52 km$^2$. After verification with satellite maps, it was found to have extremely high precision and accuracy (Figure 2).

2.3.2. UGS Data

As an important component of this study, we chose the data from "Chengdu Park Urban Green Space System Planning (2019–2035)" as the basis for the UGS space division (http://www.chengdu.gov.cn/ (accessed on 20 July 2022)). By sorting out the UGSs in the urban built-up areas and adjacent areas of Chengdu that are "free", "readily accessible", and "have leisure and tourism functions" (UGSs that are opposite to the above description, such as paid UGSs that only have ecological protection functions, are not considered in this study). Then, considering the densely populated urban built-up area of Chengdu and the high degree of land development, we included small UGSs with sightseeing functions in the study to further improve the accuracy of supply and demand evaluation [39]. Ultimately, we identified 271 UGSs with a total area of 146.89 km$^2$.

In other aspects, in traditional supply research, the geometric center of UGS is often used to represent the entrance in accessibility analysis. To make the model more suitable for real urban environments, we obtained the gate coordinate data for each UGS based on a combination of POI data, online maps, and field surveys. In UGSs with no gates or fully open borders, the relevant road intersection was considered the entrance (Figure 3a). Subsequently, based on differences in the scale, recreational services, and travel modes of UGSs, these spaces were divided into daily use UGSs and UGSs used on weekends and holidays (WH) to facilitate the subsequent calculation of the accessibility of differentiated travel modes (Table 1).

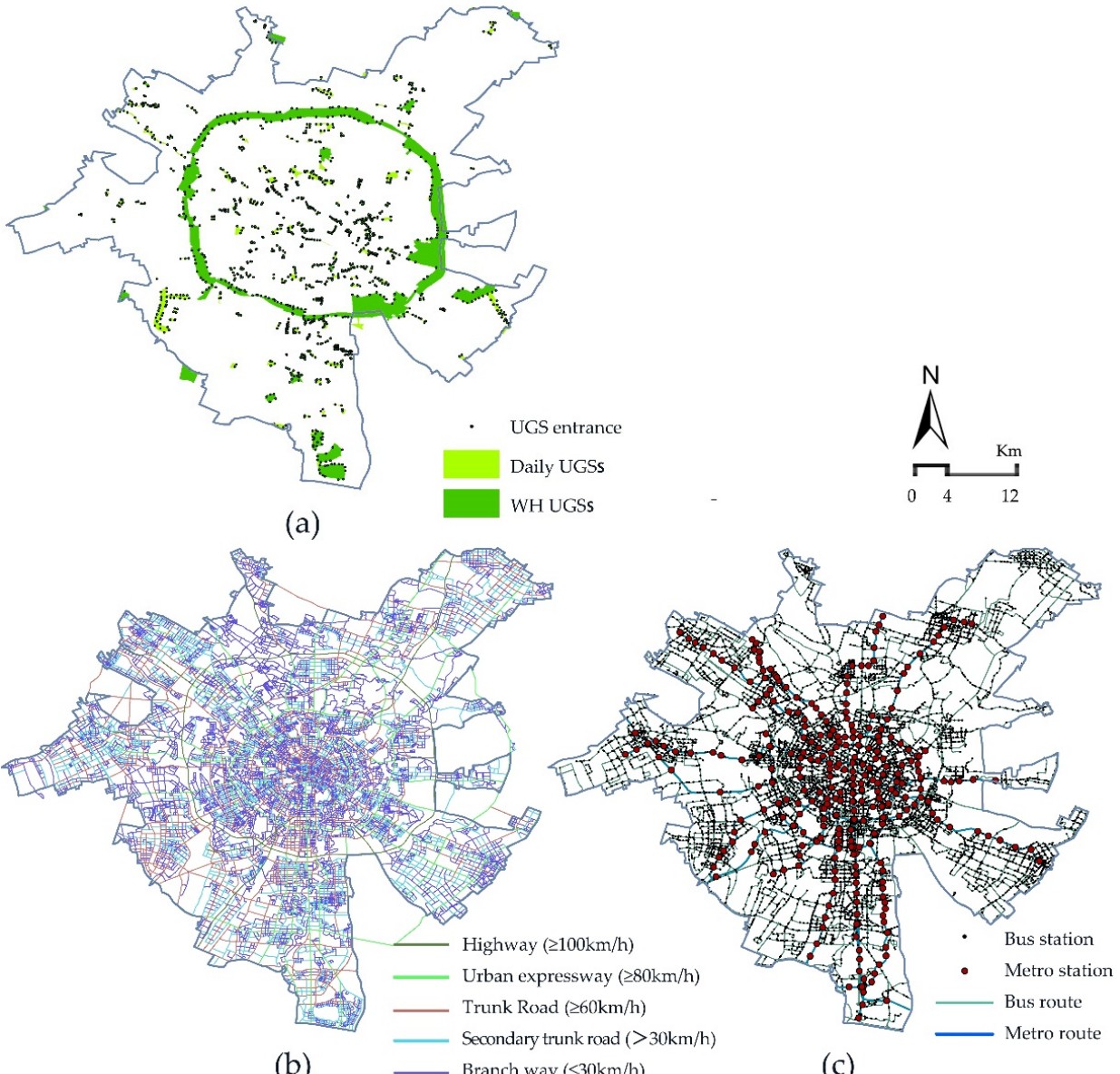

**Figure 3.** (**a**) UGSs in urban built-up areas of Chengdu; (**b**) Graded road network in urban built-up areas of Chengdu; (**c**) Public transport in urban built-up areas of Chengdu.

**Table 1.** UGS classification statistics.

| Type | Quantity/Quantity Ratio | Area (km²)/Area Ratio |
|---|---|---|
| Daily UGSs | 238/87.82% | 32.28/21.97% |
| WH UGSs | 33/12.18% | 114.61/78.03% |

### 2.3.3. Urban Traffic Data

Urban traffic data are the basis of fairness research. Based on urban traffic data from the Open Street Map (OSM) website (https://www.openstreetmap.org/ (accessed on 24 September 2022)), the road networks in built-up areas of Chengdu were extracted. However, due to the incomplete descriptions of urban branch roads in the OSM dataset, we used online map data to correct and supplement the road information. Then, roads were classified into 5 levels according to the relevant regulations of urban road construction. Due to the irregular shape of the boundaries of urban built-up areas, to ensure sufficient connections between roads and UGSs [40], we added all available information for the main roads adjacent to UGS boundaries. Additionally, to best reflect the real situation in the city, we used street view image data, adopted a combination of machine learning and artificial visual recognition, and focused on identifying road speed limit signs to obtain travel cost data suitable for the accessibility model. Finally, the Topology tool in ArcGIS 10.7 was used to conduct topological checks of roads, and the road networks in the urban built-up areas of Chengdu were obtained (Figure 3b). Specifically, urban public transport data were obtained from AmAP (https://lbs.amap.com/ (accessed on 5 January 2023)). Based on these data, we constructed a bus and subway network within the research scope (Figure 3c).

### 2.3.4. Dataset of UGS Demand

1. Dataset for objective demand of UGSs

This item contains two pieces of data. Among them, urban development demand is mainly extracted from the requirements of policies and planning documents, such as "green at 300 m and garden at 500 m", to reflect the regional characteristics of urban development. On the other hand, the service capability of UGS refers to the number of objects it needs to serve within a certain range [41]. The more objects UGS serves, the greater the demand in the region. Therefore, we choose the indicators of social demand to represent the square service objects of UGSs. In terms of data selection, studies have shown that the ratio of the number of POIs to their area can be used as an important measure of UGS service capability [41,42]. In addition, second-hand house prices in residential datasets are sensitive to changes in the surrounding environment and provide wide coverage for multiple housing types; therefore, housing price and residential population data can be used as the basis for assessing the UGS structure and social equity [43]. As a direct representation of human activities, a heatmap reflects the utilization rate of space and can confirm the vitality of space [44]. We selected three data sources, POI data, residential population data, and heatmaps, to measure the measure of social demand. The data source, processing process, and visualization method can be found in the appendix (see details from Appendix A Table A1).

2. UGS subjective demand of datasets

This content is used to analyze the subjective needs of urban residents for UGS. Previous studies often used interviews/questionnaires to obtain data [45]. This method not only takes a long time but also has a certain weakness in the amount of data. Based on this, we choose a new method to analyze the subjective demands of urban residents by using online review data. In terms of data selection, we selected all review data of green space in urban built-up areas of Chengdu before September 2023. The data source, processing process, and visualization method can be found in the appendix (see details from Appendix A Table A1).

### 2.4. Methodology and Research Process

### 2.4.1. Accessibility Model of UGSs

Currently, the research methods involving UGS accessibility mainly include the two-step floating catchment area method (2SFCA), cost-weighted distance method, and network analysis method [14,46]. Among them, the network analysis method is based on the road network, and it can be used to accurately simulate the process of residents entering UGSs;

the accessibility analysis results are accurate and objective [47,48]. The method consists of 5 elements: links, nodes, stops, centers, and turns. These elements represent possible paths from one location to another. In this study, a link refers to an urban road, a node refers to a road intersection, a center refers to the entrance/exit of a UGS (starting point for accessibility analysis), and a stop refers to the time cost associated with reaching this UGS (Figure 4a).

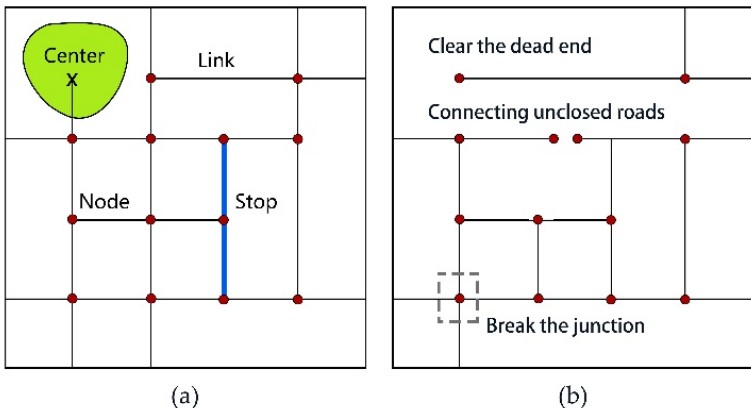

(a)  (b)

**Figure 4.** (**a**) Network analysis diagram; (**b**) topology check and road network arrangement.

For stop elements, we first set different speeds for four modes of travel: walking, cycling, public transport, and driving [49,50]. Second, a time threshold of 5–30 min was set according to the travel time acceptable to urban residents and the requirements of urban green space system development [49,51]. Additionally, considering the urban road network density and traffic congestion, a 60 s waiting time at road intersections was set uniformly. In terms of transportation to travel destinations, walking is the main means for people to travel short distances, so reaching daily UGSs by walking is easy; for WH UGSs, cycling, public transport, and driving were mainly adopted (Table 2). Then, we conducted topology checks, performed network arrangement, and constructed a network dataset (Figure 4b). Finally, the accessibility of UGSs for different travel modes was calculated, and reclassification was performed in ArcGIS 10.7 to assign weights to calculate comprehensive accessibility.

**Table 2.** Travel mode and resistance setting.

| Travel Mode | Time Threshold (min) | Speed (m/min) | For the UGS Type |
|---|---|---|---|
| Walking | 5, 10, 15 | 80 | Daily UGS |
| Cycling | 5, 10, 15, 30 | 200 | all |
| Public transportation | 5, 10, 15, 30 | The subway is 1300, and the bus is 800 | all |
| Motor vehicle | 5, 10, 15, 30 | Set according to the road speed limit | WH UGS |

### 2.4.2. Quantification of UGS Demand Dataset

1. Quantification of social demand in objective demand

To evaluate the objective needs, we used POI, residential population, and heatmap data as the basis. Kernel density estimation (KDE) in ArcGIS 10.7 was used for the quantitative assessment of POI and residential population data [44]. Notably, POI scores were calculated based on the density of information points. The residential population data were based on the number of households reported and the ratio of 2.49 persons/households in the latest census data. The formula is as follows:

$$f(s) = \sum_{i=1}^{n} \frac{1}{h^2} k \left( \frac{s - c_i}{h} \right) \tag{1}$$

where $f_{(s)}$ is the kernel density function in space $s$, $h$ is the distance attenuation threshold, $n$ is the number of elements with a distance from position $s$ less than or equal to $h$, $k$ is the spatial weight function, and $c_i$ is a core element.

The mean value algorithm was used to extract the mean value of the urban population distribution from heatmap data at the weekly scale [44,52]. The formula is as follows:

$$\overline{W} = \frac{\sum \overline{H}}{n}$$
$$\overline{H} = \frac{\sum H_{ix}}{24} \tag{2}$$

where $\overline{H}$ is the daily average population density in unit i, $H_{ix}$ is the population density in unit i at time x, $\overline{W}$ is the weekly average population distribution, and $n$ is the number of days with data.

Finally, after the three datasets were normalized in ArcGIS, the quantified social demand data were obtained by means of weighted superposition.

2. Quantification of subjective demand

Based on the online review data of Ctrip UGS, we crawled the online review data of 271 UGSs and ultimately obtained 64,234 review texts from 102 UGSs with review information. Then, a preliminary screening was performed, and duplicates were removed from the review text. Importing Text Data into ROST_CM 6 software completes word segmentation operations and manually removes meaningless entries such as "conjunctions", "place nouns", and "adverbs", completing the preliminary cleaning of comment texts.

The demand content is often highly correlated with the focus and negative emotions in the review text [53]. The focus can be intuitively understood through word frequency sorting. However, due to the separation of adjectives and nouns after word segmentation, our search for the content of this requirement can be divided into the following two steps. First, we choose to search for negative tourist perception vocabulary. Second, we searched for sentences containing negative vocabulary and analyzed their content. Based on word semantics and previous research, the results are further screened and classified into three categories: ecological demand, recreation demand, and convenience demand [54] (Table 3).

**Table 3.** Classification of Online Review Data.

| Classification | Focus of Attention | Negative Emotions |
|---|---|---|
| Ecological demand | Greening, trees, sculptures, water quality, squares, etc. | There are so few trees that they are too exposed to sunlight, dancing too much, making too much noise, etc. |
| Recreation demand | Leisure, walking, dancing, fitness, running, etc. | Lack of children's activities, fitness facilities, etc. |
| Convenience demand | Free of charge, public transportation, subway, surrounding facilities, etc. | Inconvenient transportation, limited surrounding facilities, etc. |

Finally, the intensity of various requirements in each UGS was calculated. The formula is as follows:

$$D_i = \frac{n_i}{N} \tag{3}$$

where i represents the type of demand, $n_i$ represents the number of occurrences of class *i* demand, N represents the total number of UGS comment texts corresponding to the demand, and $D_i$ is the intensity of a certain type of demand in a certain UGS.

However, due to the differences in the number of review texts among different UGSs, this difference in data magnitude makes it impossible to compare the results between different UGSs. Therefore, we used the min–max normalization method to normalize all data [55]. The formula is as follows:

$$\text{Direct indicator}: K_{\text{i}} = \frac{(X_{\text{i}} - X_{\min})}{(X_{\max} - X_{\min})}$$
$$\text{Contrary indicator}: K_{\text{i}} = \frac{(X_{\max} - X_{\text{i}})}{(X_{\max} - X_{\min})} \tag{4}$$

where $X_{\text{i}}$ is the measured value of index i, $X_{\max}$ is the maximum value of index i, $X_{\min}$ is the minimum value of index i, and $K_{\text{i}}$ is the standardized value of the indicator.

### 2.4.3. Fairness Analysis of UGSs

The Gini index and Lorenz curve analysis methods are often used as quantitative indicators of social equity performance evaluation, which can effectively understand spatial equity at the index level [56,57]. To reflect the matching status of population size and urban UGS accessibility in Chengdu, we selected the residential population data from Chengdu and studied the results of UGS accessibility. The Gini index and Lorenz curve were used to calculate the gaps in the spatial distribution of UGS. The formula applied is as follows:

$$G = 1 - \sum_{i=1}^{n} (P_i - P_{i-1}) \times (S_i + S_{i+1}) \tag{5}$$

where $G$ is the Gini coefficient, which is in the range of [0, 1]; $P_i$ is the cumulative proportion of the population; and $S_i$ is the cumulative proportion of the accessibility score.

The larger the $G$ value is, the more uneven the distribution of UGSs in the study area. According to the mathematical meaning of the Gini index, when $G \leq 0.2$, the UGS distribution is highly homogeneous; when $0.2 < G \leq 0.4$, the distribution of UGSs is reasonable and relatively homogeneous; and when $0.4 < G$, the spatial distribution gap of UGSs is expansive, with an observable lack of fairness in terms of UGS accessibility

### 2.4.4. Location Assignment Model for UGSs

The optimal allocation of UGSs requires that the supply and demand of UGSs match [58,59]. To solve this problem, we refer to Cooper's location assignment model (LA model) [60]. Based on the LA model, we determined the service dead zone of UGS supply and the mismatch area of supply and demand and then superimposed the two results. In addition, the Fishnet tool in ArcGIS 10.7 was used to construct the possible points (supply points) of UGSs based on a 100 × 100 m grid and the superposition results, and we considered an ideal situation in which all residents required access to UGSs. The residential data in these areas were used to establish the demand points for UGSs. The decision functions for supply points and demand points are as follows:

$$Y_n \in (0,1), n \in N$$
$$X_{mn} \in (0,1), m \in M, n \in N \tag{6}$$

where $m$ is the number of demand points, $M$ is the set of demand points, $n$ indicates the number of UGS supply points, and $N$ indicates the set of UGS supply points. $Y_n$ is the decision variable for the available supply points. When $Y_n = 0$, no UGS is created at position n, and vice versa when $Y_n = 1$. $X_{mn}$ represents the relationship between demand point m and UGS supply point $n$. When $X_{mn} = 0$, it indicates that the newly added UGS supply point n does not provide UGS services for demand point m, and when $X_{mn} = 1$, the opposite is true.

When determining the new UGS points, considering the layout efficiency and the constraint conditions in the aforementioned accessibility analysis, resistance levels were set. Then, these values were combined with the existing position allocation layer data and settings, and the position allocation model in ArcGIS 10.7 was used to obtain the optimal solution set for the selected optimization objective. The corresponding formulas are as follows:

$$S = \sum_{m \in M} \sum_{n \in N} D_{mn} X_{mn} \tag{7}$$

$$P = \text{s.t.} \sum_{n \in N} Y\text{n}$$
$$D_0 \geq D_{mn} X_{mn} \tag{8}$$

$S$ is the objective function set obtained with location optimization, and $D_{mn}$ is the travel cost from a demand point to a supply point. $P$ is a newly planned UGS supply point in the region to be optimized. $D_0$ is the maximum travel cost threshold from an optimized point to a newly added UGS supply point.

## 3. Results

### 3.1. Supply and Demand Evaluation of UGS in Chengdu

3.1.1. Supply Evaluation of UGS

1.　Accessibility Analysis of Different Travel Modes

According to the classification of travel modes and UGSs in the second part of this study, accessibility analysis of different travel modes was completed (see details in Appendix B Figure A1). Due to the dense UGSs in the centers of built-up areas in Chengdu, the accessibility results mainly corresponded to walking in the urban built-up areas within the time threshold of 15 min, with only a few service dead zones near the edges of the urban built-up areas. However, within 10 min of walking, only 20.27% of UGSs were accessible, indicating generally poor accessibility (Table 4). However, the accessibility of other travel modes can fully cover the urban built-up area of Chengdu within 30 min. Among them, the coverage of cycling, public transport, and driving reached 27.07%, 35.18%, and 22.14%, respectively, within 5 min, and the accessibility was better than that of walking.

**Table 4.** Cumulative service area statistics for accessibility based on different modes of travel.

| Travel Mode | Travel Time and Cumulative Service Area (km$^2$)/Ratio | | | |
| --- | --- | --- | --- | --- |
| | 5 min | 10 min | 15 min | 30 min |
| Walking | 128.53/8.23% | 316.52/20.27% | 1391.22/89.09% | - |
| Cycling | 422.67/27.07% | 836.35/53.56% | 1126.19/72.12% | 1481.91/94.90% |
| Public transportation | 549.3/35.18% | 993.1/63.60% | 1236.03/79.16% | 1548.93/99.19% |
| Motor vehicle | 345.74/22.14% | 673.73/43.15% | 1030.1/65.97% | 1533.1/98.18% |

2.　UGS supply evaluation and service dead zone analysis

To understand the supply of UGSs in urban built-up areas of Chengdu, we conducted a weighted overlay on the accessibility results of different travel modes. Based on the relevant literature [13,50,61,62] and expert consultation (a total of 16 expert-related fields were surveyed), the walking-oriented development mode in today's society was considered. The weights were calculated as follows: 0.529 for walking, 0.317 for cycling, 0.126 for public transportation, and 0.028 for motor vehicles (CI = 0.0649). In the end, the UGS supply results were reclassified from levels 0–3, representing service dead zones, low-coverage areas, generally high-coverage areas, and high-coverage areas for accessibility (Figure 5).

Through UGS supply analysis, we find that the UGS supply in urban built-up areas of Chengdu is good at present, but there are significant differences in the spatial distribution, and the overall performance is "decreasing from the center to the outside". A total of 83.25% of the urban built-up area covered accessible UGSs, but there was still an imbalance among regions (Table 5). Specifically, the areas with high accessibility account for 22.98% of the overall area and are mainly concentrated in the centers of urban built-up areas and the surrounding districts and counties with high building density. The spatial layout of the UGSs in these areas is relatively optimal, and accessibility is high. However, there are service dead zones in the western, eastern, and northeastern parts of the region, which account for an area of 358.88 km$^2$, indicating that people living in these areas cannot enjoy UGS services fairly and that the supply of urban built-up areas in Chengdu needs to be improved.

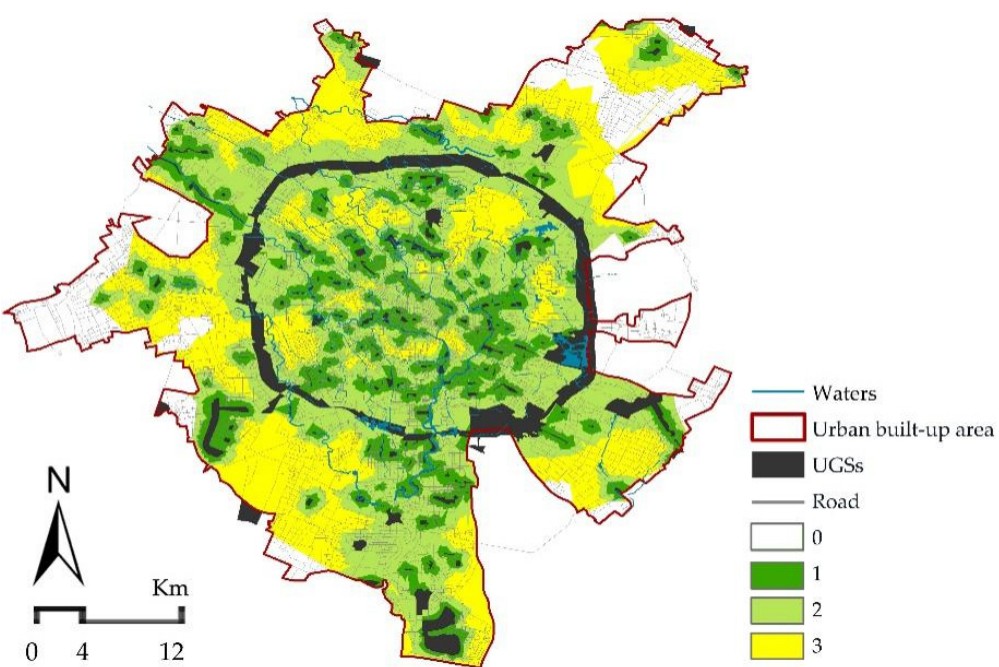

**Figure 5.** Evaluation of UGS Supply in Chengdu.

**Table 5.** Aggregate service area statistics regarding UGS supply.

| Grade | Classification | Cumulative Service Area (km$^2$)/Ratio |
|---|---|---|
| 0 | Service dead zone | 261.6/16.75% |
| 1 | High-coverage area | 358.88/22.98% |
| 2 | General-coverage area | 1034.69/66.26% |
| 3 | Low-coverage area | 1299.92/83.25% |

3.1.2. Demand Evaluation for UGSs

1.  Objective demand of UGS

The quantification of objective requirements with Formulas (1) and (2) was obtained. Through empowerment, the superposition of four objective requirements was completed. Among them, the relative importance of heatmap and POI data in social demand is relatively high, with weight values of 0.473 and 0.314, respectively, and the sum of the weight contribution exceeds 0.7. The final result is normalized by the Fuzzy Membership tool in ArcGIS and classified into five levels. The smaller the value is, the higher the demand level (Figure 6a). In general, the objective demand of urban built-up areas in Chengdu is strong, and the proportion of high-demand areas with a score below 0.57 reached 16.29%. The spatial distribution of these areas shows a high degree of consistency with the accessibility results; that is, the space demand in the center of the urban built-up area and the surrounding counties with higher building densities is higher. Combined with the preliminary quantitative results of objective requirements (see details in Appendix B Figure A2), we found that these high-demand areas were precisely the locations of the space with relatively mature UGS development but extremely high population density, reflecting the lack of per capita UGS resources in some areas.

2.  Subjective demand for UGSs

Through Formulas (3) and (4), we completed the quantification and normalization of subjective requirements (see details in Appendix B Figure A3). We found that the three types of subjective demands have obvious spatial differences, among which ecological demand and recreation demand are strong in urban centers, while convenience demand is the opposite. The weight values of 0.323 (ecological demand), 0.419 (recreation demand), and 0.258 (convenience demand) are assigned to the three types of subjective demand and

superimposed (Figure 6b). The results show that the area of highly subjective demand with a score below 0.53 accounts for 21.17%, and the overall demand degree is higher. The subjective demand in the space is characterized by the center gathering and extending to the south and the scattered distribution at the northern and western edges.

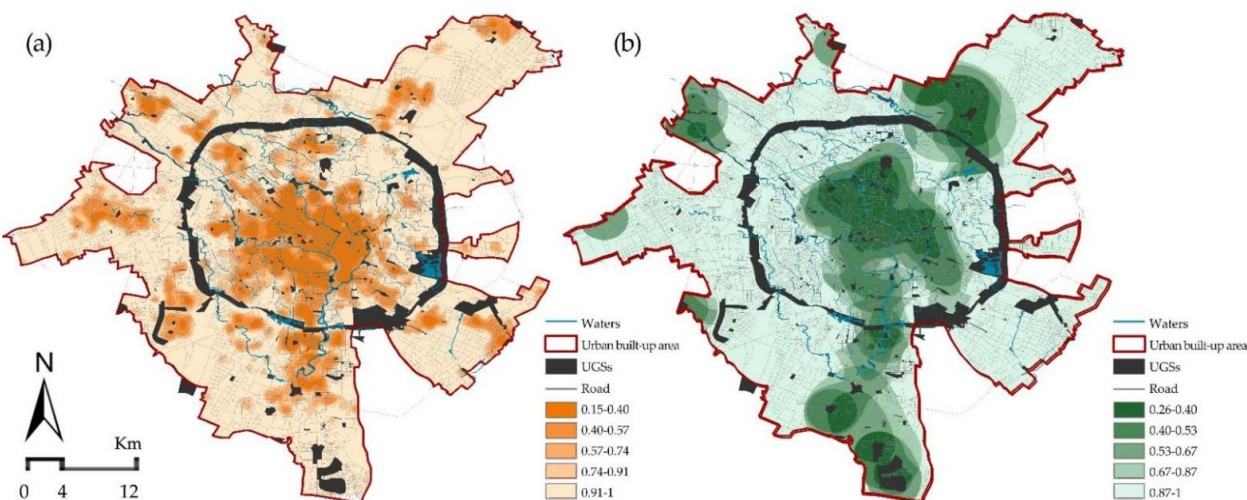

**Figure 6.** (**a**) Objective demand evaluation in Chengdu; (**b**) Subjective demand evaluation of Chengdu.

*3.2. UGS Fairness Assessment in Chengdu*

3.2.1. Gini Index and Lorenz Curve Analysis

By analyzing the fairness of UGS allocation and Formula (5), we found that the Gini coefficient between regional accessibility and population resource allocation was 0.31, so the spatial distribution of UGSs in urban built-up areas in Chengdu was reasonable and average. The characteristics of the Lorenz curve were further analyzed (Figure 7). Overall, there was a certain distance between UGS accessibility in the urban built-up areas of Chengdu and the distribution of population resources plotted on the absolute average line, reflecting a certain imbalance; that is, there are some spatial differences among UGSs, and different proportions of the population enjoy different proportions of UGS resources. From the perspective of the population with access to UGS services, 15.56% of the residents with low UGS services cannot enjoy these services. Additionally, only 2.23% of UGS resources are available to 20.05% of the cumulative population. However, approximately 15.39% of the population with high UGS services enjoy 37.86% of UGS resources.

3.2.2. Fairness Evaluation of UGS from the Perspective of Supply and Demand

The comprehensive result of demand evaluation is superimposed with the result of supply, and the value is assigned according to grades 1–3. The higher the value is, the greater the mismatch between supply and demand (Figure 8a). In general, the supply and demand of UGSs in urban built-up areas of Chengdu showed a relatively scattered distribution in space, especially in the southwest. Although the density of UGSs is high in these areas, due to the large population, there is less space available for per capita allocation, reflecting the insufficient supply of UGS space. In terms of type area proportion, the area of the supply and demand evaluation score of two accounted for the largest proportion, which was 46.29%. Second, the area with a score of three occupies 187.27 km$^2$ of urban built-up area space (accounting for 11.99%), which can be used as the focus of future location optimization of UGSs.

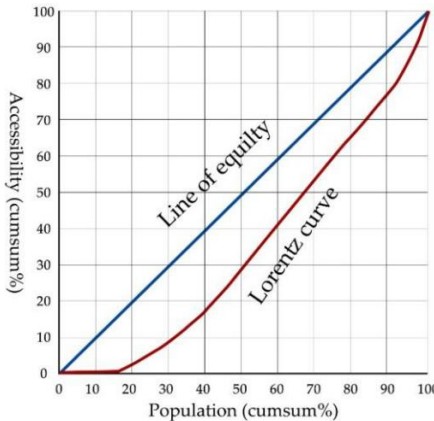

**Figure 7.** UGS accessibility and population plot with a Lorentz Curve.

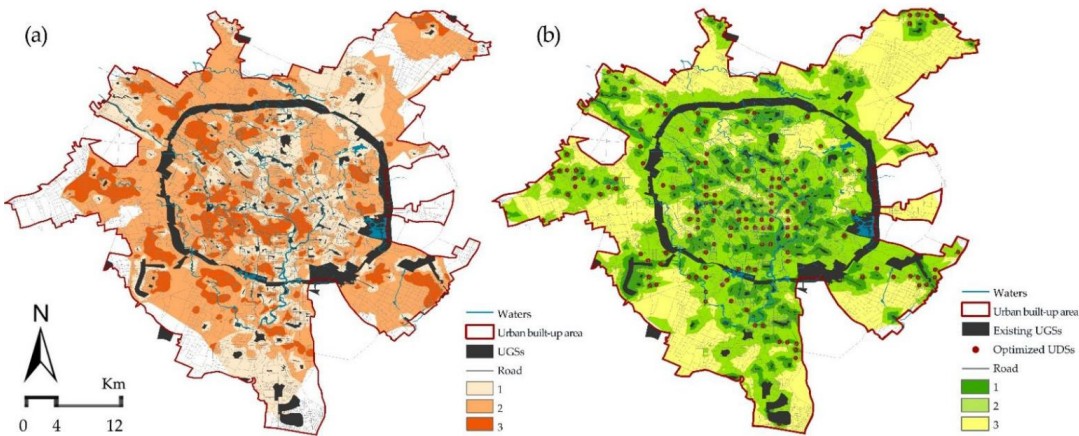

**Figure 8.** (**a**) UGS supply and demand evaluation in Chengdu; (**b**) Accessibility of UGSs based on optimized site selection in Chengdu.

### 3.3. UGS Location Optimization in Chengdu

Through Formulas (6)–(8), two allocation methods, namely, the minimum impedance model and maximum coverage area, are most commonly used in urban public facility allocation [63–66]. Furthermore, position fine-tuning was carried out through satellite image-assisted identification of open space, and 142 newly added UGSs were finally selected (Figure 8b).

The selected UGS points were mainly distributed in regions with low supply levels or blind supply and regions with unbalanced supply and demand in central China. Overall, the addition of UGS effectively increases the range of accessibility. Among them, the high-coverage area increased by 90.69 km$^2$, and the low-coverage area increased the most, with an increase of 176.63 km$^2$ (Table 6). After optimization, the Gini coefficient of UGS in urban built-up areas of Chengdu is 0.21, which is close to the high average line, which further narrows the gap in the proportion of service area and service population of all levels of UGS accessibility and greatly improves the fairness of UGS.

**Table 6.** Cumulative service area statistics of UGSs after optimization.

| Grade | Classification | Cumulative Service Area (Km$^2$)/Ratio | |
|---|---|---|---|
| | | **After Optimization** | **Before Contrast Optimization** |
| 0 | Service dead zone | 84.97/5.44% | −176.63 |
| 1 | High-coverage area | 449.57/28.79% | 90.69 |
| 2 | General-coverage area | 1171.33/75.01% | 136.64 |
| 3 | Low-coverage area | 1476.55/94.56% | 176.63 |

## 4. Discussion

### 4.1. Contribution of the Improved UGS Supply and Demand Evaluation Model to UGS Equity Research

At present, although there are many one-sided studies on the supply and demand of UGSs, the two are relatively independent, and it is difficult to directly analyze the relationship and matching between them, which is the difficulty of research in this field. Based on multisource geospatial big data, this study constructs a new perspective of UGS equity research combining supply and demand.

In terms of supply, accessibility analysis of multiple travel modes using 2SFCA and network analysis has become a major trend in this field and has become relatively mature [46,49–51]. Based on the accessibility study, we made the following two main improvements. First, we did not use the traditional method of setting road speeds based on road grading or daily experience. Based on the street view data of an online map, this paper uses the combination of machine recognition and manual visual assistance to identify the speed limit signs and simulate the real speed limit of urban roads. Second, through POI data, more accurate UGS entrances and exits are identified. The improvement of the above method can move the supply model closer to the real urban space and provide refined ideas for the future study of space supply.

In terms of demand, traditional UGS demand mainly clarifies users' subjective demand for UGSs by exploring satisfaction, cognitive differences, preferences, and other factors [8,34,67]. However, this kind of research is relatively one-sided in scale and object and cannot reflect all the needs of space. Therefore, we propose a new perspective of needs evaluation combining subjective and objective factors. At the level of objective needs, it is different from the calculation of population within administrative divisions using census data in the past [23]. We introduce residential population data and heatmap data to promote the development of social demand to a refined scale. At the subjective demand level, we introduce online review data and interpret urban residents' demand for UGSs from a functional perspective. Compared with the subjective demand, this method has the advantages of acquiring data quantity, convenient operation, and general use.

In view of previous research, supply and demand are relatively independent pain points [24,25]. In this study, the ArcGIS platform was used to quantify the data, and the research object, scale, and evaluation content were unified. This helps measure the supply and demand matching of UGSs in spatial terms rather than just in exponential dimensions. Compared with the existing research, our research results can better identify the areas of supply and demand imbalance from space and can more effectively guide the optimal layout of UGSs [68,69]. Our research can be applied not only to the planning of UGS, but also to other urban elements to realize the optimal allocation of space resources and the sustainable development of the city.

### 4.2. Reasons for the Unfairness of UGS in Chengdu

This study confirmed the evidence of the unfairness distribution of UGSs in space and found two unfairness phenomena of service dead zones and mismatches between supply and demand in urban built-up areas of Chengdu. Aiming at service dead zones, the supply study of this study found that there were green space vacancies in the western, eastern, and northeastern areas of the urban built-up areas in Chengdu, accounting for 16.75%, which was consistent with other studies in Chengdu [59]. Through investigation, we found that these areas are mainly industrial areas or areas to be developed. According to the Master Plan of Chengdu City (2016–2035), in the future, most of the contaminated industrial areas within the urban built-up area of Chengdu will gradually move out of the city to achieve industrial transformation within the urban built-up area. Therefore, these areas can also be the focus of future UGS layout and optimization.

In view of the mismatch between supply and demand, our research results show that in the central area of Chengdu, the matching results vary greatly in a small space and are relatively scattered. This also leads to a small number of people in some areas enjoying

a large amount of UGS resources. The mismatch between UGS supply and demand is particularly prominent in urban centers, which is consistent with previous research results. This is the problem that most cities will have at the present stage of development; that is, the density of the old city is too high, and there is not much space for the development of UGSs. As a result, the growth of UGSs, which represents the quality of urban space, cannot keep up with the growth of the population, so UGSs in urban central areas cannot meet the needs of urban residents.

### 4.3. Strategies to Promote Equity in UGS

According to the research results, there are still obvious inequities in UGSs in urban built-up areas of Chengdu, so it is imperative to improve the service efficiency of UGSs. We have made the following recommendations for the future development of the region.

First, the number of UGSs should be increased to eliminate "service dead zones" and achieve full coverage of UGS resources and services. There are still many service dead zones in the layout of UGSs in urban built-up areas of Chengdu. The main gaps are in the industrial zone and the areas to be developed. Therefore, for the industrial zone, it is suggested to use the existing space to establish banded UGSs, thus integrating recreation and protection functions. For the area to be developed, UGS planning should be performed in the appropriate space in advance to achieve a complete green space layout.

Second, UGS establishment should be based on population distribution and demand. UGSs belong to the category of public services, and the planning and construction of UGSs should be based on the population situation and should be accurately matched with the population. This research also verifies this point of view in the need assessment. Therefore, in the future development of UGSs, planning and construction should be based on population dynamics and requirements, including those of tourists; this approach will help achieve a fair distribution of UGS resources to the greatest possible extent.

Third, the allocation of UGSs of different sizes should be accounted for, and attention should be given to the development and construction of small green spaces. According to the results of this study, there is a certain imbalance between supply and demand in the Chengdu city center. At the same time, the construction content of such urban centers has been basically perfected, leaving little room for the development of new UGSs. To maximize the utilization of resources in the limited space, it is suggested to construct pocket parks and microgreen spaces by interplanting green spaces to further improve the fairness of UGS and the utilization rate of urban space.

Finally, the construction of public transportation and slow traffic systems should be strengthened. Our research found that public transport is slightly more accessible than other modes of travel. Therefore, it is suggested to add a bus stop directly to UGSs to improve the public transport system. At the same time, walking and cycling, as the most convenient and fast way to travel short distances, should strengthen the construction of urban slow traffic systems such as greenways and nonmotor vehicle lanes and set up multiple shared bicycle parking points at the entrances and exits of each UGS to ensure the efficiency and accessibility of residents' walking trips.

### 4.4. Limitations and Prospects

First, urban space is highly complex and changeable. 2SFCA, network analysis, and other traditional supply research methods are already very mature. However, these relatively static UGS supply research methods, which mostly start from the average travel speed, cannot reflect the special situation of morning and evening peak congestion and are still not accurate enough to simulate the rapidly changing urban space. Future reachability research can further improve the real-time and accuracy of reachability results through online maps and multi-time real-time reachability calculations.

Second, a refined indicator of the type of UGS supply and demand can be considered, especially on the supply side. For example, our UGS fairness study divided the types of UGS demand from the demand side but did not divide them from the supply side. Such a

pattern may result in inconsistent data structures on the supply and demand side. Future research may consider exploring the fairness of UGSs by type on both sides of supply and demand. Wang et al.'s study provides such an idea [70]. However, because UGSs have many functions and strong compounds, wetlands have ecological, leisure, and science popularization functions [71]. For this reason, the classification of types in their research is subjective and not sufficiently accurate. Follow-up studies could enable more rigorous and comprehensive UGS feature identification.

Third, the quantification of subjective demand has always been difficult in supply–demand-matching research, and many scholars have tried to solve this problem through questionnaires, socioeconomic data, and social media data [72]. However, for large-scale urban space research, a questionnaire survey is a heavy workload, and it is difficult to accurately describe the actual needs of urban residents. This study chooses online review data to try to solve this problem. This kind of data relies on a certain geographical space point (UGS, public facilities, places, etc.) and its service radius to represent. Although the large amount of data can make up for the problem of coverage, it may still lead to the risk of service dead zones. At the same time, the people covered by online review data are usually middle-aged or young people, and it is difficult to cover the needs of the elderly and children [73]. Therefore, which scale and data to choose for a more accurate and comprehensive subjective demand analysis can still be the focus of future research.

## 5. Conclusions

This study discusses the fairness of UGS in urban built-up areas of Chengdu from the perspective of supply and demand, improves the theoretical framework of UGS fairness evaluation by combining supply and demand, and provides a method model that can be extended. First, based on multisource geospatial big data, this study analyzed the supply of UGSs with accessibility analysis and evaluated the demand for UGSs from subjective and objective perspectives. Second, the fairness of UGS was analyzed from the two aspects of coefficient and space. Finally, UGS fairness was optimized based on the LA model. The improved supply and demand evaluation model clarifies the fairness evaluation method of UGS, measures the spatial fairness of urban built-up areas, and provides methodological ideas and strategic guidance for the improvement of UGS in Chengdu.

According to the empirical research results of Chengdu, we found that the travel mode directly affects the supply range of UGSs and is reflected in fairness. At the same time, in the case of a highly dense population, the UGS of the city is in short supply; that is, the equity is negatively correlated with the population.

In future work, a more comprehensive fairness evaluation system and more real-time, refined, and diverse data and methods need to be built to provide support for fairness research to achieve accurate identification and optimization of UGS fairness.

**Author Contributions:** Conceptualization, Q.D.; methodology, X.L.; software, Q.D.; validation, H.L., X.L. and L.W.; investigation, Q.D. and L.W.; resources, H.L.; data curation, Q.D.; writing—original draft preparation, Q.D.; writing—review and editing, Q.D. and P.H.; visualization, X.L. and D.L.; supervision, H.L. and P.H.; project administration, X.L., Y.W. and X.C. All authors have read and agreed to the published version of the manuscript.

**Funding:** This research was funded by the Xihua University talent introduction program, grant number w2320048.

**Institutional Review Board Statement:** Not applicable.

**Informed Consent Statement:** Not applicable.

**Data Availability Statement:** The data are not publicly available due to privacy.

**Conflicts of Interest:** The authors declare no conflict of interest.

## Appendix A

**Table A1.** Objective demand dataset and preprocessing.

| Type | POI Data | Residential Population Data | Heatmap Data | Online Review Data |
|---|---|---|---|---|
| Acquisition mode | AmAP (https://lbs.amap.com/ (accessed on 3 January 2023)) | Anjuke (https://chengdu.anjuke.com (accessed on 25 December 2022)) | Baidu Maps (https://lbsyun.baidu.com/ (accessed on 23 and 26 November 2022)) | Ctrip (https://you.ctrip.com/ (accessed on 29 August 2023)) |
| Information covered | Includes information such as name, classification, address, latitude and longitude, etc. | Contains the name of the neighborhood, coordinates, housing price, number of residential units, etc. | Precise location of the phone user | Includes review text, rating, etc. |
| Function | Reflect urban functional facilities | Reactive residential population | Reflect the distribution and aggregation of people | Reflect the perception and preference of urban residents |
| Pretreatment method | Manual screening based on spatial calibration and data cleaning | | - | Text analysis |
| Processing result | 600,130 POI records | 12,273 residential area records | The weekly average of the population distribution | A total of 64,234 comments were collected and 118,435 words were selected |
| Quantization mode | KDE analysis in ArcGIS | | Mean value algorithm | Natural breaks |

## Appendix B

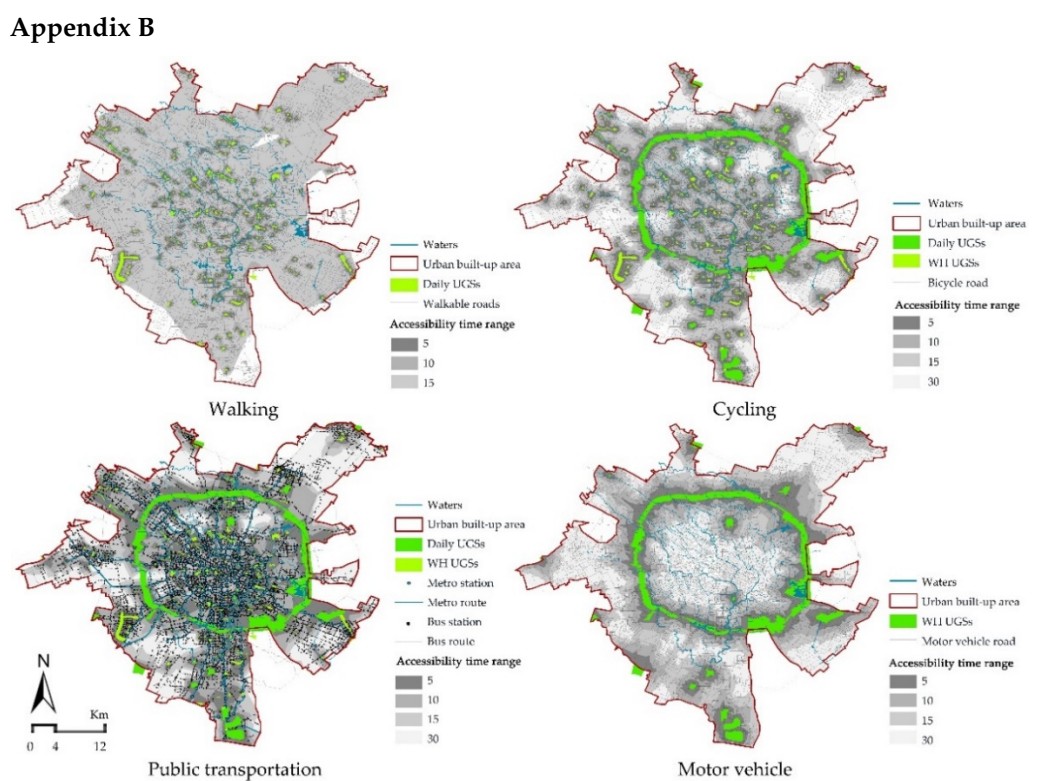

**Figure A1.** Accessibility of UGSs in Chengdu based on 4 travel modes.

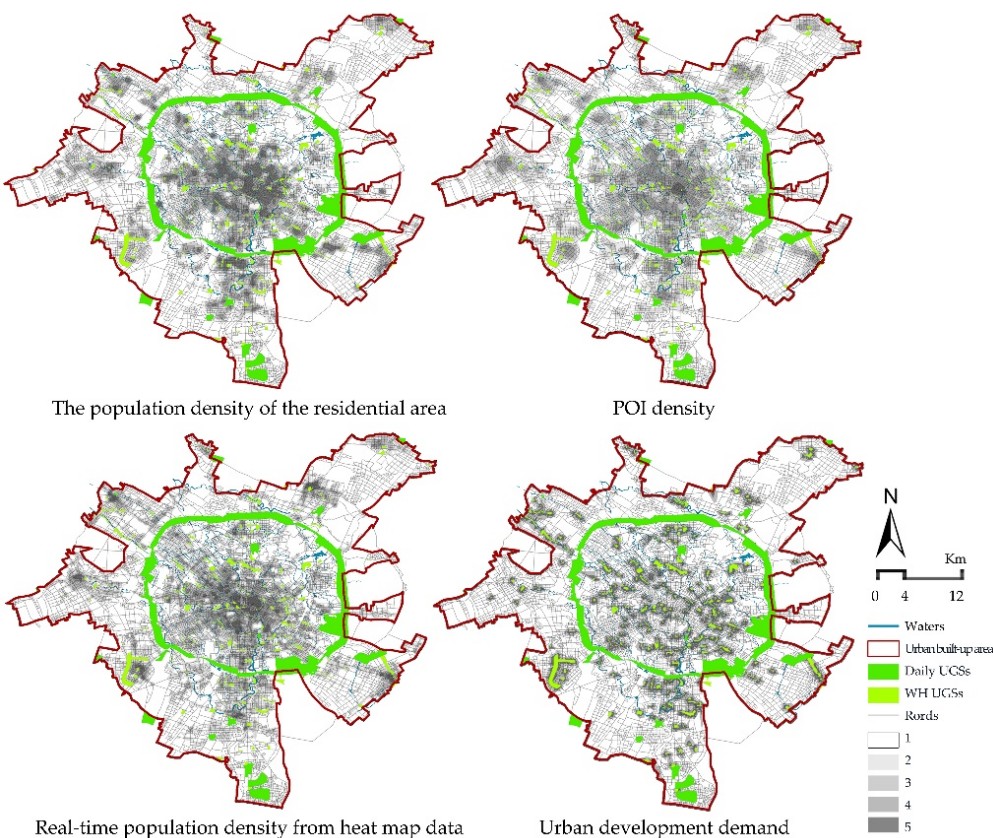

The population density of the residential area

POI density

Real-time population density from heat map data

Urban development demand

**Figure A2.** Quantitative processing of objective demand.

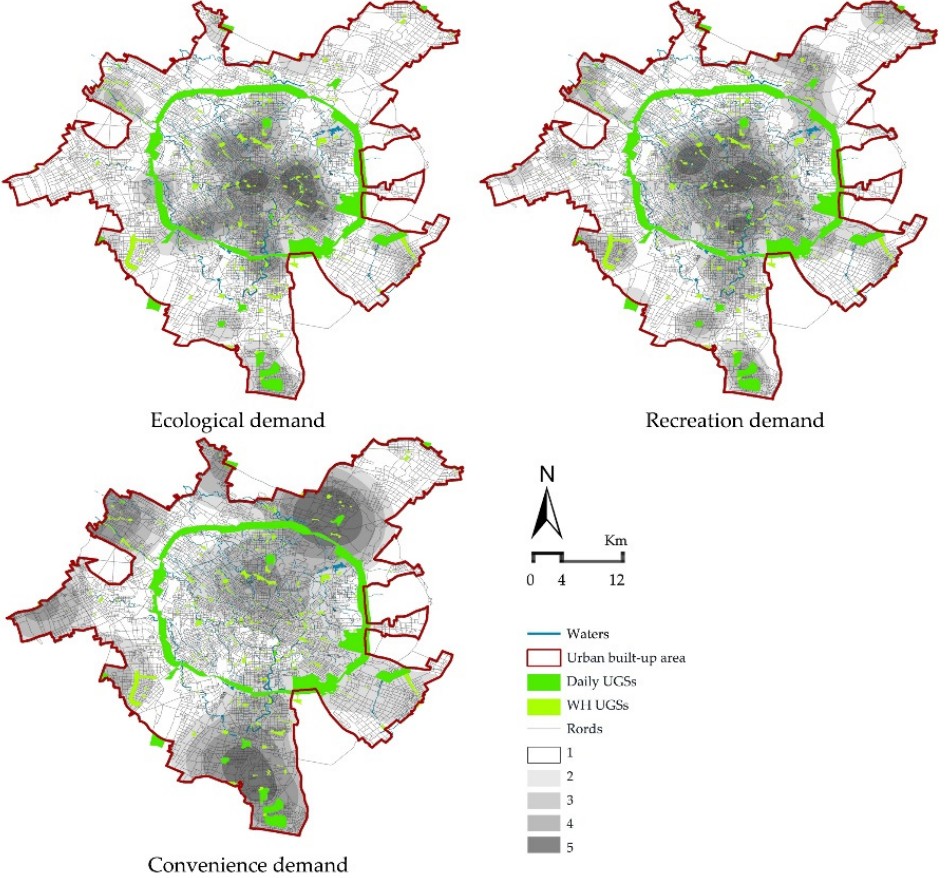

Ecological demand

Recreation demand

Convenience demand

**Figure A3.** Quantitative processing of subjective demand.

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
