# Peer review of "Evaluation and Optimization of Green Space Fairness in Urban Built-Up Areas Based on an Improved Supply and Demand Model: A Case Study of Chengdu, China"

_sustainability, doi:10.3390/su152015014_

Round 1

Reviewer 1 Report (Previous Reviewer 1)

The paper deals with an interesting topic and is well laid out,  some corrections in detail:     - 1 Introduction.
  • In lines 42-43 Replace "matching", "restricts" and "enjoyed by" with " relationship",  "limits" and "for residents".
  • In lines 49-52 replace the phrase with the clearer one: "Therefore, it is necessary to develop a more scientific method of equality research that promotes the equal use of urban infrastructure by urban residents and contributes to the promotion of the health and well-being of urban resident.";
  • In line 68 replace "compensate" with "make up" and "defect" wih "shortcoming";
  • In lines 70 replace the phrase with the clearer one: "Some researchers have combined modes of transportation (e.g., cars, public transit, pedestrians, and bicyclists), comprehensively considered the accessibility of UGS to different modes of transportation, and included differences in transportation choices for different groups of people";
  • In line 79 replace "the division(..)" with "classification according to road quality or focused on a single travel mode dimension, resultings in evaluation results that are inconsistent with the actual situation of the city.";
  • In line 82 replace "Interpreted (...)" with "explained from the supply side of single accessibility, and the explanation of the service availability of UGSs is also weak."
  • In lines 91-94 replace the phrase with a clearer one: " Relying on the quantitative data of recreation, ecological regulation and carbon footprint provided by UGS, some scientists have proposed a key assessment system to determine the supply and demand of UGS.";
  • In line 105 replace "grades" with "quality";
  • In line 109 replace "real" with "actual";
  • In lines 103-104 replace the phrase with a clearer one: " This content is an important basis for defining the boundaries of our research and completing data processing.";
3- Results
  • In line 422 replace "accesabilit" with "accesability";
  • In line 483 "ly"

Appendix B

  • Make the figures clearer and more readable

The bibliography is not very 'international'. You could add, for example:

  • Which urban agriculture conditions enable or constrain sustainable food production?
  • (2023) International Journal of Agricultural Sustainability

  • Exploring rooftop rainwater harvesting potential for food production in urban areas

    Lupia, F.Baiocchi, V.Lelo, K.Pulighe, G. Agriculture (Switzerland)2017, 7(6), 46
  •  
  • Combined Effects of Climate and Pests on Fig (Ficus carica L.) Yield in a Mediterranean Region: Implications for Sustainable Agricultural Strategies
  • (2023) Sustainability (Switzerland)

Author Response

Point 1: The paper deals with an interesting topic and is well laid out,  some corrections in detail:

-1 Introduction.

In lines 42-43 Replace "matching", "restricts" and "enjoyed by" with " relationship",  "limits" and "for residents".

In lines 49-52 replace the phrase with the clearer one: "Therefore, it is necessary to develop a more scientific method of equality research that promotes the equal use of urban infrastructure by urban residents and contributes to the promotion of the health and well-being of urban resident.";

In line 68 replace "compensate" with "make up" and "defect" wih "shortcoming";

In lines 70 replace the phrase with the clearer one: "Some researchers have combined modes of transportation (e.g., cars, public transit, pedestrians, and bicyclists), comprehensively considered the accessibility of UGS to different modes of transportation, and included differences in transportation choices for different groups of people";

In line 79 replace "the division(..)" with "classification according to road quality or focused on a single travel mode dimension, resultings in evaluation results that are inconsistent with the actual situation of the city.";

In line 82 replace "Interpreted (...)" with "explained from the supply side of single accessibility, and the explanation of the service availability of UGSs is also weak."

In lines 91-94 replace the phrase with a clearer one: " Relying on the quantitative data of recreation, ecological regulation and carbon footprint provided by UGS, some scientists have proposed a key assessment system to determine the supply and demand of UGS.";

In line 105 replace "grades" with "quality";

In line 109 replace "real" with "actual";

In lines 103-104 replace the phrase with a clearer one: " This content is an important basis for defining the boundaries of our research and completing data processing.";

3- Results

In line 422 replace "accesabilit" with "accesability";

In line 483 "ly"

Response 1: We appreciate your careful reading of our research, as well as language suggestions. According to your careful suggestion, we have modified the corresponding content.

Point 2: 

Appendix B

Make the figures clearer and more readable

Response 2:  Thanks for your suggestions, we have improved the legibility of this form by adjusting the line spacing and so on.

Point 3: 

The bibliography is not very 'international'. You could add, for example:

  • Which urban agriculture conditions enable or constrain sustainable food production?
  • Combined Effects of Climate and Pests on Fig (Ficus carica L.) Yield in a Mediterranean Region: Implications for Sustainable Agricultural Strategies

Response 3: Thank you for your suggestions on the references, we have replaced some of the references and supplemented the references you suggested, which is very useful to us.

Reviewer 2 Report (Previous Reviewer 3)

The manuscript investigates the fairness of UGS from the perspective of supply and demand and improves the theoretical framework of supply and demand combination evaluation. Its structure is well-organized, and the study's purposes, research questions, hypothesis, and methods are clearly stated. However, a brief explanation of the study's data and methodology should be written in the Introduction. In addition, although the study is a local case study, the methodology and the importance of the obtained results for other study areas anywhere in the world should be discussed. Furthermore, the concept of "geospatial big data" should be explained. In this context, the critical question is: Are the data used in the study big data? All in all, the manuscript can be published in the journal after minor revisions.

Author Response

Point 1:However, a brief explanation of the study's data and methodology should be written in the Introduction.

Response 1: Thank you for your suggestion. We have added in lines 122-125 to achieve a complete representation of the data and methods.

Point 2:In addition, although the study is a local case study,  the methodology and the importance of the obtained results for other study areas anywhere in the world should be  discussed.

Response 2: Thank you for your suggestion. We add in lines 561-563 to enhance the breadth and worldwide applicability of the study.

Point 3:Furthermore, the concept of "geospatial big data" should be explained. In this context,  the critical question is: Are the data used in the study big data?

Response 3: Thank you for your suggestion. This concept is really not well known, so we add it briefly in line 122. It should be noted that this data is actually a type of big data, indicating that big data with geospatial properties. Because of its openness, easy access, and large amount of data, it still belongs to the category of open data in big data.

Reviewer 3 Report (Previous Reviewer 4)

I have read the revisions. The quality of paper has improved. 

Author Response

Thank you for your recognition of our research, and thank you again for your valuable comments on our research before.

Reviewer 4 Report (Previous Reviewer 5)

None, the changes have improved the manuscript for publication.

Author Response

Thank you for your recognition of our research, and thank you again for your valuable comments on our research before.

This manuscript is a resubmission of an earlier submission. The following is a list of the peer review reports and author responses from that submission.

Round 1

Reviewer 1 Report

The work is interesting but some things need to be better explained and some aspects need to be improved for the paper to be accepted as a scientific communication:

1) The state of the art part needs to be expanded, as it is too small this way

2) You should better explain what you mean here: "They used high-resolution (30 m) global artificial impervious area (GAIA) data, 123 which can depict the extent of urban built-up areas well", what kind of model is this?

3) In the next line, it makes no sense in such a large area to report all those decimal places, and here and afterwards the "two" of the square kilometres must always be as an exponent in a scientific article.

4) The maps in fig.2 are too small and unreadable, you should enlarge them, possibly to a higher resolution.

5) If you mention software (e.g. ArcGIS) you should put the version and specification of the modules used

6) In equation 1 you should put space or the multiplication sign between the two terms in brackets 

7) The wording of formulae 3 and 4 is inverted and not positioned correctly

8) Formula 5 is missing, if it is the one above 6 you must insert a blank line between the two

9) Can you better specify the ArcGIS 'Fishnet tool' you are talking about

10) The following equations again have incorrect numbering and the first one is not clear

11) Figures 5 and 8 are also too small

12) When you say: "analysis using an ArcGIS network analysis method." You need to be more specific

Author Response

Sustainability

July 20, 2023

Sustainability-2448789

Dear editors and reviewers,

On behalf of all of the coauthors, we would like to thank all of you for your critical comments and helpful suggestions concerning our original manuscript entitled “Accessibility, service pressure measurement and location optimization of high-density urban green space”. The comments were all very valuable for revising and improving the quality of our manuscript as well as for providing guidance regarding our future work. We studied the comments carefully and revised the manuscript accordingly, which we hope will be met with your approval and will satisfy the standards of the journal. Our responses to the comments are provided below, as is a summary of the main revisions to the manuscript.

We hope that the revisions and accompanying responses make our manuscript suitable for publication in Sustainability. We look forward to hearing from you at your earliest convenience.

Sincerely,

Qidi Dong

  • mail: dqd@mail.xhu.edu.cn

--------------------------------------------------------------------------------------------------------------The following are our point-by-point responses to your constructive comments.

--------------------------------------------------------------------------------------------------------------

Response to Reviewer 1’s Comments

Point 1: The state of the art part needs to be expanded, as it is too small this way

Response 1: Thank you for your constructive comment. Based on your question, we are not quite sure what "The state of the art part" refers to. Therefore, we enlarged all the images that were too small.

Point 2: You should better explain what you mean here: "They used high-resolution (30 m) global artificial impervious area (GAIA) data, 123 which can depict the extent of urban built-up areas well", what kind of model is this?

Response 2: Thank you for your suggestion. This content is the construction principle for the selected urban built-up area data. Perhaps we were not clear enough, so we have rewritten the content.

Point 3: In the next line, it makes no sense in such a large area to report all those decimal places, and here and afterwards the "two" of the square kilometres must always be as an exponent in a scientific article.

Response 3: Thank you for your suggestion. We have made adjustments to the number units in the full text.

Point 4: The maps in fig.2 are too small and unreadable, you should enlarge them, possibly to a higher resolution.

Response 4: Thank you. We have enlarged all the images that were too small.

Point 5: If you mention software (e.g. ArcGIS) you should put the version and specification of the modules used

Response 5: Thank you for your suggestion. We have added the corresponding software information.

Point 6: In equation 1 you should put space or the multiplication sign between the two terms in brackets

Point 7: The wording of formulae 3 and 4 is inverted and not positioned correctly

Point 8: Formula 5 is missing, if it is the one above 6 you must insert a blank line between the two

Point 9: The following equations again have incorrect numbering and the first one is not clear

Responses 6-9: Due to our negligence, the formula numbers were wrong. We are very sorry for the trouble. Thank you for your thoughtful advice. We have updated the formula numbers and added corresponding symbols for clarity.

Point 10: Can you better specify the ArcGIS 'Fishnet tool' you are talking about

Point 11: When you say: "analysis using an ArcGIS network analysis method." You need to be more specific

Responses 10-11: The Fishnet tool in ArcGIS is a tool used to build a square grid. The supply points of UGS cannot be accurately determined in the urban space based on the preliminary data analysis. Therefore, we used the 'Fishnet tool' to build grid points to help determine the locations of the preselected UGSs (Data Management Tools→Feature Class→Create Fishnet in ArcGIS).

Point 11: Figures 5 and 8 are also too small

Response 11: Thank you. We have enlarged all the images that were too small.

Again, we are grateful for your positive comments and suggestions.

Reviewer 2 Report

Thank you for an interesting manuscript. The following suggestions are large and small: 

* As the paper covers Chengdu, please specify Chengdu, China in the title, and emphasize Chengdu in the abstract, introduction and conclusion. 

* The paper would be much better if the authors wrote and thought more deeply and critically about the impact of their study. What are the practical implications? What are the key takeaways? What are the barriers? What are the limitations? What can and should be done differently?  What can be applied elsewhere? This may not sound like much but truly engaging with these questions will be a lot of work. 

Minor copyediting needed. 

Author Response

Sustainability

September 20, 2023

Sustainability-2448789

Dear editors and reviewers,

On behalf of all of the coauthors, we would like to thank all of you for your critical comments and helpful suggestions concerning our original manuscript entitled “Accessibility, service pressure measurement and location optimization of high-density urban green space”. The comments were all very valuable for revising and improving the quality of our manuscript as well as for providing guidance regarding our future work. We studied the comments carefully and revised the manuscript accordingly, which we hope will be met with your approval and will satisfy the standards of the journal. Our responses to the comments are provided below, as is a summary of the main revisions to the manuscript.

We hope that the revisions and accompanying responses make our manuscript suitable for publication in Sustainability. We look forward to hearing from you at your earliest convenience.

Sincerely,

Qidi Dong

  • mail: dqd@mail.xhu.edu.cn

--------------------------------------------------------------------------------------------------------------The following are our point-by-point responses to your constructive comments.

--------------------------------------------------------------------------------------------------------------

Response to Reviewer 2’s Comments

Point 1:As the paper covers Chengdu, please specify Chengdu, China in the title, and emphasize Chengdu in the abstract, introduction and conclusion.

Response 1: Thank you for your suggestion. Our study focuses on the city of Chengdu, so we have further emphasized the scope of the study in the title, introduction and conclusion in accordance with your suggestion.

Point 2: The paper would be much better if the authors wrote and thought more deeply and critically about the impact of their study. What are the practical implications? What are the key takeaways? What are the barriers? What are the limitations? What can and should be done differently?  What can be applied elsewhere? This may not sound like much but truly engaging with these questions will be a lot of work.

Response 2: Thank you for your constructive comment. Our study does lack an in-depth explanation of the impact and significance of the research results. Therefore, according to your suggestions, we have made special additions to the introduction, discussion and conclusion to better reflect the significance of our research work.

Again, we are grateful for your positive comments and suggestions.

Reviewer 3 Report

The reviewed study, of which novelty and scientific soundness are low, investigated accessibility, service pressure measurement, and location optimization of high-density urban green spaces (UGS) in Chengdu, China. Unfortunately, some critical points prevented it from being published in the journal: The study's main purpose and research questions were not clearly defined, and the importance was not clearly stated. It should have emphasized why the topic is important. How to relate to current knowledge? What are the differences from previous studies? What is the main objective of the study? Besides, the literature review is too weak and was not written according to an international perspective; thus, it can be interpreted as a local study.

Furthermore, the Discussion section should be rewritten because the obtained results should have been discussed in this section. However, it was written like a literature review. As for the Conclusion section, it should be improved and written from a global perspective. My further remarks are as follows:

  • The authors mentioned "...single dimensions" in line 18 and "... geospatial big data" in lines 18-19. The terms should be explained in the text.
  • The citation style should be as in the Instruction for Authors: "reference numbers should be placed in square brackets [ ], and placed before the punctuation; for example [1], [1–3] or [1,3]."
  • Figure and table numbers should be checked (for ex. line 306)
  • The river systems should be represented as a line in the figures.
  • The sentence in lines 120-123 cannot be understood.
  • In lines 233-236, a citation is needed for the sentence.
  • What is the "t" in Equation 6?
  • In line 327: Were the consistency rates of expert consultations checked, how to select the experts, and what are the other methods?

Author Response

Sustainability

September 20, 2023

Sustainability-2448789

Dear editors and reviewers,

On behalf of all of the coauthors, we would like to thank all of you for your critical comments and helpful suggestions concerning our original manuscript entitled “Accessibility, service pressure measurement and location optimization of high-density urban green space”. The comments were all very valuable for revising and improving the quality of our manuscript as well as for providing guidance regarding our future work. We studied the comments carefully and revised the manuscript accordingly, which we hope will be met with your approval and will satisfy the standards of the journal. Our responses to the comments are provided below, as is a summary of the main revisions to the manuscript.

We hope that the revisions and accompanying responses make our manuscript suitable for publication in Sustainability. We look forward to hearing from you at your earliest convenience.

Sincerely,

Qidi Dong

  • mail: dqd@mail.xhu.edu.cn

--------------------------------------------------------------------------------------------------------------The following are our point-by-point responses to your constructive comments.

--------------------------------------------------------------------------------------------------------------

Response to Reviewer 3’s Comments

Point 1: The authors mentioned "... single dimensions" in line 18 and "...  geospatial big data" in lines 18-19.  The terms should be explained in the text.

Response 1:  Thank you. In this study, "... "single dimensions" means" single dimensions". For clarity, we replaced this phrase with "single dimensions". Notably, geospatial big data is a type of big data and has now become a proper term. We have added explanations in the main text.

Point 2: The citation style should be as in the Instruction for Authors: "reference numbers should be placed in square brackets [ ], and placed before the punctuation;  for example [1], [1–3] or [1,3]."

Response 2: Thank you for your careful suggestion. We have adjusted the specifications of the full references.

Point 3: Figure and table numbers should be checked (for ex. line 306)

Response 3: Thank you for your careful suggestion. We have proofread the numbers and continuity of the figures, tables and formulas in the full paper.

Point 4: The river systems should be represented as a line in the figures.

Response 4: Thank you for your constructive comment. Due to the large spatial scale of the study and the complexity of the elements in the map, it is difficult to see the river clearly. Therefore, we adjusted and enlarged the illustration and simultaneously adjusted the river in the legend as a line representation.

Point 5: The sentence in lines 120-123 cannot be understood.

Response 5: Thank you. Notably, lines 120-123 represent the source and significance of data for urban built-up areas. We rewrote this part of the paper for clarity.

Point 6: In lines 233-236, a citation is needed for the sentence.

Response 6: Thank you for your suggestion. We give a description of why we chose the Gini index and Lorenz curve in our research in Chengdu. This is a description of our research process, so there is no need for a reference. It may be that our language was unclear, so we made changes to avoid ambiguity.

Point 7: What is the "t" in Equation 6?

Response 7: Here, s.t. in Equation 6 is an abbreviation for "subject to" and represents constraints. This may be due to the absence of a "t" after the "t" in our formula. To avoid confusion, the text has been updated.

Point 8: In line 327: Were the consistency rates of expert consultations checked, how to select the experts, and what are the other methods?

Response 8: Thank you for your constructive comment. We omitted many steps in this process, and according to your suggestion, we have added the number of experts and CI value (consistency) for clarity.

Point 9: For other questions about review, discussion, etc.

Response 9: Thank you for your suggestion. According to your suggestions, we have rewritten the research review, discussion and other parts of the paper. The comparison from an international perspective is strengthened in the review section. Due to the requirements of the journal for the discussion section, namely, "The findings and their implications should be discussed in the broadest context possible. Future research directions may also be highlighted”, we limited the evaluation of our own results in this study. According to your suggestion, we reduced the discussion section and compared the results of this study with those of other studies.

Point 10:International issues of perspective

Response 10: Thank you for your constructive comment. There are other reviewers who said, "As the paper covers Chengdu, please specify Chengdu, China in the title, and emphasize Chengdu in the abstract, introduction and conclusion." Therefore, we have made corresponding modifications to focus on the research scope. As the conclusion you proposed should be improved from a global perspective, we have made the following reflections and improvements: 1. strengthened the international perspective of the review and the international applicability of the research objectives, and 2. in the conclusion, emphasized that this study can be applied broadly in the field of UGS optimization.

Again, we are grateful for your positive comments and suggestions.

Reviewer 4 Report

This paper is concerned with accessibility and location optimization of the UGSs in the urban built-up areas of Chengdu, China. The topic is an important and research on UGSs in developing countries is generally lacking.

The paper is good in terms of methodology and analysis. However, the theoretical aspect needs to improve. I have several comments that may contribute to improving quality of paper as follows:

1.     The literature on studied phenomenon is limited, particularly, international practices and studies in terms of UGS. For example, UGS is one of the key factors of Sustainable Development Goals (SDGs), particularly Goal 11. It is important to explain the importance of this study in the context of SDGs.   

2.     Authors have presented factors affecting accessibility to UGS including the number, location, distance, area, shape, health, and religious attributes. But they did not discuss the scope of these factors in social, economic, and environmental dimensions. Highlighting this scope is significant both theoretically and practically, where it allows following up and comparing accessibility to UGS between cities of China.

3.     Authors selected Chengdu, the capital of Sichuan Province, as a case study, without justifying the selection of this city. Justifications should be provided.

4.     One of the interesting results of this study is that public transport system in Chengdu is complete and capable of providing convenient travel methods. Authors need to provide interpretations about this result. For instance, is it related to smart mobility? Or is there strong progress on urban sustainability indicators in terms of both UGS and public transport?    

Author Response

Sustainability

September 20, 2023

Sustainability-2448789

Dear editors and reviewers,

On behalf of all of the coauthors, we would like to thank all of you for your critical comments and helpful suggestions concerning our original manuscript entitled “Accessibility, service pressure measurement and location optimization of high-density urban green space”. The comments were all very valuable for revising and improving the quality of our manuscript as well as for providing guidance regarding our future work. We studied the comments carefully and revised the manuscript accordingly, which we hope will be met with your approval and will satisfy the standards of the journal. Our responses to the comments are provided below, as is a summary of the main revisions to the manuscript.

We hope that the revisions and accompanying responses make our manuscript suitable for publication in Sustainability. We look forward to hearing from you at your earliest convenience.

Sincerely,

Qidi Dong

  • mail: dqd@mail.xhu.edu.cn

--------------------------------------------------------------------------------------------------------------The following are our point-by-point responses to your constructive comments.

--------------------------------------------------------------------------------------------------------------

Response to Reviewer 4’s Comments

Point 1: The literature on studied phenomenon is limited, particularly, international practices and studies in terms of UGS.  For example, UGS is one of the key factors of Sustainable Development Goals (SDGs), particularly Goal 11.  It is important to explain the importance of this study in the context of SDGs.

Point 2: Authors have presented factors affecting accessibility to UGS including the number, location, distance, area, shape, health, and religious attributes.  But they did not discuss the scope of these factors in social, economic, and environmental dimensions.  Highlighting this scope is significant both theoretically and practically, where it allows following up and comparing accessibility to UGS between cities of China.

Responses 1-2: Thank you for your constructive comments. Our research on the previous theory and review is not extensive enough, and the content is relatively limited. According to your suggestions, we have reviewed and expanded this part of the paper to give our research theoretical and practical significance. In view of various economic, social and environmental factors, this issue is also the focus of equity research. However, since most of China's socioeconomic data are presented as statistical data with large ranges, they are suitable for large-scale research at urban agglomeration level. When applied to our study, there would be a mismatch in data scale, so we did not consider this type of comparison.

Point 3: Authors selected Chengdu, the capital of Sichuan Province, as a case study, without justifying the selection of this city.  Justifications should be provided.

Response 3: Thank you for your suggestion. According to your comment, we added "2.1. Study area" to strengthen the significance and reason for choosing Chengdu city.

Point 4: One of the interesting results of this study is that public transport system in Chengdu is complete and capable of providing convenient travel methods.  Authors need to provide interpretations about this result.  For instance, is it related to smart mobility?  Or is there strong progress on urban sustainability indicators in terms of both UGS and public transport?

Response 4: Thank you. This is a very interesting idea. Our team also considered this topic when writing the paper. However, unfortunately, according to the journal writing template, the results section “should provide a concise and precise description of the experimental results, their interpretation, as well as the experimental conclusions that can be drawn." Therefore, limit the extended analysis in this chapter. Additionally, this content is not the main focus of this research. All in all, it is not quite clear where this content should go. Based on the above reasons, we carried out a certain expansion analysis at the end of 3.1. If we have not added enough relevant content, please let us know in the next review, and we will think about where we can add more.

Again, we are grateful for your positive comments and suggestions.

Reviewer 5 Report

Generally, I consider the article quite relevant in terms of spatial equity. Regardless the specific context in which the methodology has been applied, I consider applicable to very many contexts worldwide. The type of data, the software packages and the methodologies are fairly available in many cities. Therefore, this work provides sufficient and sound contributions for sustainable urban planning to be considered for publication.

Just a few pieces of advice for slight improvement of the article:

1. Figure 2b: road hierarchies should be better distinguished with different colors, not only blues

2. Figure 2c: bus station is not clearly distinguished

3. Table 1: what is PCS?

4. Table 1: better differentiate ratios in separate columns

5. Table 2: are these max speeds? this could bias the results, shouldn't it be average at working hours?, or please discuss it/clarify it

6. Table 2: for public transportation: why not only WH?

7. Line 251: wording needs improvement

8. Section 2.3.4.: is the assumption that all households will be needing the UGSs ? could this be challenged? please provide clarification or discuss it

9. section 3.1. / point 1: could you provide modal distribution of mobility? that would help to better interpret the results

English was clear, regardless of some typing errors in figure 9 (optimized GUS???) and wording (line 251). 

Author Response

Sustainability

September 20, 2023

Sustainability-2448789

Dear editors and reviewers,

On behalf of all of the coauthors, we would like to thank all of you for your critical comments and helpful suggestions concerning our original manuscript entitled “Accessibility, service pressure measurement and location optimization of high-density urban green space”. The comments were all very valuable for revising and improving the quality of our manuscript as well as for providing guidance regarding our future work. We studied the comments carefully and revised the manuscript accordingly, which we hope will be met with your approval and will satisfy the standards of the journal. Our responses to the comments are provided below, as is a summary of the main revisions to the manuscript.

We hope that the revisions and accompanying responses make our manuscript suitable for publication in Sustainability. We look forward to hearing from you at your earliest convenience.

Sincerely,

Qidi Dong

  • mail: dqd@mail.xhu.edu.cn

--------------------------------------------------------------------------------------------------------------The following are our point-by-point responses to your constructive comments.

--------------------------------------------------------------------------------------------------------------

Response to Reviewer 5’s Comments

Point 1: Figure 2b: road hierarchies should be better distinguished with different colors, not only blues

Point 2: Figure 2c: bus station is not clearly distinguished

Responses 1-2: Thank you for your constructive comment. For the road and public transportation illustrations in Figure 2, similar colors and too small a legend made it difficult to read. We have changed these figures for clarity.

Point 3: Table 1: what is PCS?

Point 4: Table 1: better differentiate ratios in separate columns

Responses 3-4: Thank you for your suggestions. First, PCS is a unit of quantity. Perhaps our expression was wrong, so it was deleted. Additionally, we added definitions of the quantity ratio and area ratio for clarity.

Point 5: Table 2: are these max speeds? this could bias the results, should not it be average at working hours? , or please discuss it/clarify it

Point 6: Table 2: for public transportation: why not only WH?

Responses 5-6: Thank you. This speed is the average speed of the tenderer, which may not have been clear in the previous paper. Therefore, we provided supplementary explanations to the corresponding content. The relation between UGS type and public transportation is strong because public transportation in Chengdu is relatively convenient. In our early research, many urban residents (especially elderly individuals, for whom public transportation is free in China) were surveyed. Additionally, most residents were willing to reach UGSs by public transport each day. In view of these factors, we have provided supplementary information in the corresponding parts of the paper.

Point 7: Line 251: wording needs improvement

Response 7: Thank you for your suggestion. The 'bandwidth' in line 251 is difficult to effectively define. It represents the bandwidth in ArcGIS, and we have removed this text to avoid ambiguity.

Point 8: Section 2.3.4. : is the assumption that all households will be needing the UGSs ?  could this be challenged?  please provide clarification or discuss it

Response 8: Thank you. In this study, we set up an idealized scenario in which it is assumed that all residents require access to UGSs Another explanation is that in new areas of the city (without residents), urban planners usually use the distribution of green space as a means of planning in the early stage. This approach is common, and it aligns with most of the UGS blind spots found in our study. The idea you proposed is also very interesting. Through an investigation of residents in blind spot areas, the demand of residents for UGS can be determined, and such an analysis could aid in funding UGSs. Additionally, supply and demand would be well matched. However, due to time constraints, we could not complete the survey of the needs of residents in the whole city. We explain this limitation in section 2.3.4.

Point 9: section 3.1.  / point 1: could you provide modal distribution of mobility?  that would help to better interpret the results

Response 9: Thank you for your suggestion. We are not quite sure what you mean by "mobility". We consider it the direction of travel of the crowd in the accessibility analysis. Is this correct? To address this lack of clarity, we added a reachability analysis and corresponding discussion in section 2.3.2.

Again, we are grateful for your positive comments and suggestions.
